# Prevalence and correlates of contraceptive use among adolescent mothers: Results from a cross-sectional survey in Ouagadougou, Burkina Faso, and Blantyre, Malawi

Emmanuel Oloche Otukpa[1]*, Alister Munthali[2‡], Nathalie Sawadogo[3‡],
Boniface Ayanbekongshie Ushie[4], Anthony Idowu Ajayi[1]

1 African Population and Health Research Center, Manga Close, Nairobi, Kenya, 2 Centre for Social Research, University of Malawi, Zomba, Malawi, 3 Institut Superieur Des Sciences De la Population, Universite Joseph Ki-Zerbo, Ouagadougou, Burkina Faso, 4 Beshi King Development Services, Abuja, Nigeria

◉ These authors contributed equally to this work.
‡ AM and NS also contributed equally to this work
* eotukpa@aphrc.org

## Abstract

### Background

Adolescent motherhood remains a pressing public health challenge in sub-Saharan Africa, where repeat pregnancies exacerbate the risks of maternal morbidity, disrupted education, and entrenched economic vulnerability. Contraceptive use among sexually active adolescent mothers offers a critical pathway to mitigate these adverse outcomes, yet empirical evidence on prevalence, determinants, and barriers to contraceptive uptake within this population remains strikingly limited across the region. Our study examines contraceptive use, its correlates, and reasons for non-use of contraception among adolescent mothers (aged 12–19) in Ouagadougou, Burkina Faso and Blantyre, Malawi.

### Methodology

The data analyzed for this study were part of a larger cross-sectional survey on the lived experiences of pregnant and parenting girls. For this study, we limited our analysis to 628 (Ouagadougou) and 500 (Blantyre) adolescent mothers who were neither pregnant nor trying to get pregnant. We randomly selected urban and rural enumeration areas (EAs) in the study settings, conducted household listing, and identified eligible participants whom we interviewed. We used interviewer-administered questionnaires to obtain information on contraceptive awareness, use, and reason for non-use, and analyzed the data using descriptive and inferential statistics.

**Data availability statement:** All associated dataset files are available from the APHRC Microdata portal database (accession number(s) DDI-APHRC-PPA-2023-V01) at https://aphrc.org/microdata-portal/

**Funding:** This research was part of a larger study "Understanding the context of, and addressing sensitive sexual and reproductive rights issues in sub-Saharan Africa: A regional research and advocacy programme." funded through a grant to the African Population and Health Research Center from the Swedish International Development Cooperation Agency (Sida Contribution No. 12103). The funders had no role in study design, data collection and analysis, decision to publish, or preparation of the manuscript.

**Competing interests:** The authors have declared that no competing interests exist.

## Results

Knowledge of contraceptive methods is nearly universal among adolescent mothers, and the contraceptive use prevalence was higher than the national average in both study settings. Single adolescent mothers were significantly less likely to be currently using any contraceptive methods (AOR: 0.75; 95% CI: 0.61–0.93 in Burkina Faso; AOR: 0.60; 95% CI: 0.50–0.71 in Malawi) or any modern methods (AOR: 0.66; 95% CI: 0.49–0.90 in Burkina Faso; AOR: 0.55; 95% CI: 0.46–0.66 in Malawi) compared to their married counterparts in both countries. Adolescent mothers aged 19 were more likely to be current users of any contraceptives compared to those aged 16 or younger in Burkina Faso (AOR: 1.55; 95% CI: 1.07–2.26) and were also more likely to have ever used any methods in Malawi (AOR: 1.28; 95% CI: 1.07–1.53). Infrequent sex (Malawi, 64.9%; Burkina |Faso, 36.6%) was the main reason for contraceptive non-use, though fear of side-effects (17.7%) and religious prohibition (16.6%) also stood out.

## Conclusion

A substantial proportion of adolescent mothers, especially single and younger adolescents, are not using any contraceptive methods or relying on less effective methods. These results underscore the need for more interventions targeting adolescent mothers with accurate information on contraceptive methods. These interventions should address the religious objection to contraceptives in Burkina Faso and infrequent sex in Malawi.

## Introduction

Rapid repeat pregnancies pose significant health and socioeconomic risks to adolescent mothers and their children. Rapid repeat pregnancies, defined as pregnancies occurring within 18 months or less of a previous birth, are associated with increased risks of maternal morbidity and mortality, adverse birth outcomes, and intergenerational poverty [1]. When adolescent mothers face rapid, successive pregnancies, their chances of returning to school or enrolling in vocational training, essential for their economic empowerment, diminish, consequently heightening their risk of intergenerational poverty. Utilizing effective contraceptives is essential to preventing rapid repeat pregnancies and empowering adolescent mothers socioeconomically [1,2]. Expanding access to a diverse range of contraceptive methods not only empowers adolescent mothers to exercise agency over their reproductive health but also serves as a pivotal indicator of progress in reducing rapid repeat pregnancy rates [3]. Equitable availability of these methods ensures informed decision-making, addressing both biological vulnerabilities and structural barriers, such as limited healthcare access and socio-cultural stigma, that disproportionately affect this population [4,5].

Malawi and Burkina Faso have made efforts to improve contraceptive prevalence rates through various policy initiatives and health system-strengthening

measures [4,5]. In Malawi, for instance, the government has implemented programs to increase access to family planning services in both rural and urban areas, including the provision of free contraceptives and community-based outreach services [6]. Similarly, Burkina Faso has prioritized family planning within its national health strategy, with interventions aimed at increasing awareness, improving service delivery, and addressing socio-cultural barriers to contraceptive use [7].

Despite these efforts, challenges remain in achieving optimal contraceptive prevalence rates and preventing rapid repeat pregnancies among adolescent mothers in both countries. Generally in sub-Saharan Africa (SSA), short birth spacing is higher among adolescent mothers compared to older women; at 58.7% compared to 43.9% [8] In addition to the risks presented by SBIs, persistent socio-cultural norms, limited access to healthcare facilities, contraceptive commodities stock-outs, and the general reluctance of health care providers to provide and/or prescribe contraceptives to adolescent girls without parental knowledge and consent (provider bias); continue to hinder women's ability to access and use contraception effectively [9–11]. Moreover, disparities exist in contraceptive use among different demographic groups, with rural, less educated, and economically disadvantaged women often facing greater barriers to accessing family planning services [12]. Data on contraceptive knowledge and use among adolescent mothers is vital for shaping efforts to strengthen health systems, promote gender equity, empower adolescents and young women to make autonomous reproductive choices, reduce rapid repeat pregnancies, and improve maternal and child health outcomes.

In identifying the key correlates of contraceptive use among adolescent mothers, policymakers and healthcare practitioners can design targeted and culturally sensitive interventions that cater to the unique needs of populations in Burkina Faso and Malawi and subsequently limit the risk of rapid repeat pregnancies. Given the importance of current data to inform policies and programs, we analyzed data from a mixed-methods study of which examined the lived experiences of 'pregnant and parenting' adolescents. The data obtained from this study explored the contextual experiences of adolescent girls in Burkina Faso and Malawi; who were pregnant or had a child less than two years old and adolescent boys who have fathered a child. consequently, this paper is an attempt to examine contraceptive knowledge, use and correlates in Burkina Faso and Malawi.

## Materials and methods

### Study design and settings

Data was obtained from a larger mixed methods study examining the lived experiences of pregnant and parenting adolescents. Specifically, this cross-sectional descriptive study investigated the prevalence and correlates of contraceptive use among adolescent mothers in urban and rural settings in Ouagadougou, Burkina Faso, and Blantyre, Malawi. A detailed methodology has been published elsewhere [13]. In summary, the larger study involved a concurrent equal status mixed methods design, triangulating data from qualitative and quantitative approaches. Survey participants were selected using a two-stage stratified sampling approach. Overall, we surveyed a total of 980 and 669 adolescent girls in Burkina Faso and Malawi, respectively. These locations were chosen based on their distinctive socio-cultural and healthcare contexts, specifically, the high adolescent childbearing rates, allowing for a broader understanding of the factors influencing contraceptive knowledge and practices among adolescent mothers. Burkina Faso, situated in West Africa, and Malawi, located in Southeast Africa, share similarities but also manifest unique characteristics. Both countries grapple with challenges such as high fertility rates, limited healthcare infrastructure, and deeply ingrained socio-cultural beliefs and practices towards the expression of sexuality among adolescents [10]. Yet, they differ in terms of religious diversity, urbanization rates, and socioeconomic development. By conducting a comparative analysis of these two countries, this study aims to unravel the multifactorial drivers behind contraceptive prevalence and provide nuanced insights that can guide tailored interventions for reproductive health improvement.

## Study participants and sampling

This analysis focused primarily on adolescent mothers who were not pregnant at the time of the study and were not actively trying to become pregnant. The larger study initially included 980 pregnant and parenting adolescents in Burkina Faso and 669 in Malawi. However, after excluding pregnant adolescents and those actively trying to become pregnant, we were left with 628 (Burkina Faso) and 500 (Malawi) adolescent mothers for the analysis. Participants were recruited using a two-stage probability sampling strategy. In the first stage, rural and urban enumeration areas (EAs) were randomly selected, 71 EAs in Burkina Faso (29 rural EAs and 42 urban EAs) and 66 EAs in Malawi (26 rural EAs and 40 urban EAs). An enumeration area (EA) is a geographic unit used primarily for census and survey purposes. It is the smallest geographic area into which a country is divided for the collection of population and housing data. The study settings were the Central region, which includes Ouagadougou and six rural municipalities (Pabré, Loumbila, Tanghin Dassouri, Saaba, Komsilga, and Koubri) around the town in Burkina Faso, and Blantyre and environs in Southern Malawi. The lists of EAs were obtained from the national statistical offices in both countries. For the second stage, we conducted a comprehensive household listing exercise to identify pregnant and parenting adolescent girls who are residents in those EAs. We subsequently compiled a sampling frame consisting of the list of all the households with pregnant and parenting adolescents from which eligible participants in the selected households were included in the study. We determined that a sample size of 500 adolescent mothers is adequate to address the research question of this study. This conclusion is based on the formula for calculating sample size in an observational study with an infinite population, a 95% confidence interval, an error margin of ±0.43, and a prevalence of adolescent childbearing at 15%.

## Data collection

Data collection took place between 19/07/2021 to 26/09/2021 in Burkina Faso and 21/03/2021 to 07/05/2021 in Malawi. We recruited and trained research assistants in ethics for research involving human participants, the study tools, objectives and consenting procedures. The trained research assistants administered a pre-tested questionnaire face-to-face to eligible and consenting adolescent girls. The questionnaire was developed around a comprehensive literature review of previously validated tools (such as the Global Early Adolescent Study tools) [14] and adapted to the local context and language (translated into Chichewa and French). Before data collection, the tool was pilot tested to identify issues and assess appropriateness. Given the sensitive nature of the study, we developed a distress protocol highlighting how to identify distressed adolescents and procedures for referral for psychosocial support. We accessed this data from the African Population and Health Research Center (APHRC) microdata portal in 09/09/2024 for further analysis.

## Ethical considerations

This research received approval from the APHRC's Internal Scientific review board ref *No: DOR/2021/005.* as well as University of Malawi Research Ethics Committee (UNIMAREC) *Ref No:P.12/20/42* and the Burkina Faso Ethics committee for Health Research *deliberation no:2021-04-090.* We sought signed informed consent from the emancipated participants aged 19 and lower. For adolescents under the age of 18 and still living with their parents, we obtained signed minor ascent in addition to signed informed parental consent.

## Variables and measures

**Outcome variables.** The main outcome variable was *contraceptive use*. We measured this by asking adolescent mothers: "Have you ever heard/used [type] method?" and "What contraceptive method are you currently using?" – with yes or no option responses. The options specified what kind of contraceptive from a list of modern and traditional contraceptive methods. Modern contraceptive methods include male or female condoms, oral contraceptive pills, emergency contraceptive pills, injectables, intrauterine devices, and implants. Traditional methods encompassed herbs,

withdrawal, rhythm, and other non-clinical practices. We derived four variables: *'ever use of any methods'*, *'ever use of modern methods'*, *'current use of any methods'*, and *'current use of modern methods,'* to comprehensively assess contraceptive use among adolescent mothers.

**Independent variables.** We included individual, family and community levels covariates based on previous studies [13,15]. Age, marital status, religion, education, employment history, and endorsement of sexual double standards were the individual-level factors included. *Age* was measured by asking participants their age as of their last birthday; this ranged from 13 to 19. This variable was treated as a categorical variable. Marital status measured whether the participant was married or in a union or partnership; it was conceived as a categorical variable with three levels: Married or Co-habiting, Separated or Divorced, and Single. *Ever worked for pay* was a binary variable that captured whether the participant had worked for payment. The *Religion* variable measured the religious affiliation of the participants; the response options included Catholic, Protestant, Islam, Traditional, and no religion. This was later collapsed into Christianity, Islam, and Other or no religion. The Education variable measured the highest educational attainment of the participant, and it is a categorical variable with three levels: No formal education, Primary and Secondary. Finally, the *living arrangements* variable measured if the participant lived with (n)either or both parents, while the residence variable measured whether the participant lived in a rural or an urban setting.

**Statistical analysis.** We analyzed the data using Stata 15.1 [16]. Our analysis took two approaches. First, we summarized the participants' socio-demographic characteristics, knowledge, and use of contraceptives using descriptive statistics, including frequencies and percentages. Then, we conducted unadjusted bivariate analyses to examine which variables were associated with contraceptive -use. Given the high prevalence of contraceptive uptake among the study participants and to adjust the standard errors, we ran adjusted, modified Poisson regression models to examine the independent predictors of contraceptive use among adolescent mothers and reported incidence rate ratios and their 95% confidence interval.

## Results

Table 1 describes the participants' characteristics. In both countries, a large percentage of girls were aged 19: 58 percent in Burkina Faso and 40 percent in Malawi. The average age of the adolescent mothers was 18.3 in Burkina Faso and 17.9 in Malawi.

In Burkina Faso, approximately four out of five adolescent mothers (82%) reported not living with their parents, but only about 50% reported not living with their parents in Malawi. More adolescent mothers reported practicing Islam in Burkina Faso (64%) than in Malawi (8%). The reverse is true for both forms of Christianity; protestant (Malawi: 72.5% vs. Burkina Faso: 7.5%) and Catholic (Malawi 19% vs. Burkina Faso 28%).

### Descriptive findings on contraceptive awareness and use

The overall contraceptive awareness and use in Burkina Faso and Malawi are shown in Fig 1. In both countries, the awareness of contraceptives was 100%. In Burkina Faso, 89% of participants reported having ever used contraceptives, compared to 86% in Malawi. Among the girls who have used contraceptives, most have used modern contraceptive methods (BFS, 86.3%; Malawi, 84.4%) in both countries.

Most participants (70.5% in Burkina Faso vs 64.8% in Malawi) were currently using contraceptives, with a vast majority using modern methods. In Burkina Faso, 55.7% of current contraceptive users used modern methods compared to 63.0% in Malawi.

Table 2 shows the prevalence of contraceptive awareness, ever used, and current use by methods. The data indicate high awareness levels in both Malawi and Burkina Faso. The most well-known methods in both countries were male condoms (93.6% in Burkina Faso vs. 91.2% in Malawi), pills (89.2% in Burkina Faso vs. 90.8% in Malawi), and injectables

**Table 1. Socio-demographic characteristics of study participants.**

| Factors | B/Faso (N = 628) n(%) | Malawi (N = 500) n(%) |
|---|---|---|
| Residence | | |
| Urban | 393(62.6) | 305(61.0) |
| Rural | 235(37.4) | 195(39.0) |
| Age | | |
| 12-16 | 32 (4.5) | 54 (10.8) |
| 17 | 56 (8.4) | 90 (18.0) |
| 18 | 201 (32.5) | 157 (31.4) |
| 19 | 339 (54.7) | 199 (39.8) |
| Living arrangements | | |
| Not living with both parents | 516 (81.9) | 250 (49.8) |
| Lives with one parent | 43 (7.0) | 141 (28.2) |
| Live with both parents | 69 (11.1) | 109 (21.9) |
| Religion | | |
| Christianity | 235(35.9) | 456(91.1) |
| Islam | 391(63.7) | 37(7.5) |
| Traditional/Other/No religion | 2(0.4) | 7(1.4) |
| Marital Status | | |
| Married/Cohabiting | 478(76.1) | 214(42.8) |
| Separated/Divorced | 54(8.6) | 63(12.6) |
| Single | 96(15.3) | 223(44.6) |
| Education | | |
| No formal education | 178(28.3) | – |
| Primary | 150(23.9) | 325(65.0) |
| Secondary/higher[a] | 300(47.8) | 175(35.0) |
| Ever worked for pay | | |
| Yes | 231(36.8) | 149(29.8) |
| No | 397(63.2) | 351(70.2) |
| Sexual double standard views | | |
| High | 408(65.0) | 146(29.2) |
| Low | 220(35.0) | 354(70.8) |

[a]3 adolescents in Malawi and 4 in Burkina Faso attained technical education and higher

(87.1% in Burkina Faso vs. 94.9% in Malawi). The least known methods were emergency contraception (42.4% in Burkina Faso vs. 57.6% in Malawi) and withdrawal (45.3% in Burkina Faso vs. 69.2% in Malawi).

Injectables (33.6% in Burkina Faso vs 49.4% in Malawi) and the Pill (27.6% in Burkina Faso vs 11.5% in Malawi) were the most ever used contraceptive methods in both countries. On the other hand, emergency contraception (13% in Burkina Faso vs 4.1% in Malawi) and female condoms (13.4% in Burkina Faso vs 0% in Malawi) were the least used.

Adolescent mothers in both countries were predominantly using Injectables (24.7% in Burkina Faso vs 41.3% in Malawi), implants (11.9% in Burkina Faso vs 14.4% in Malawi) and male condoms (23.2% in Burkina Faso vs 3.7% in Malawi) as their current contraceptive methods.

**Correlates of contraceptive use.** Tables 3 and 4 present the results of the modified Poisson regression analyses examining the association between contraceptive use and individual, community and family characteristics in Burkina Faso and Malawi, respectively. Increasing age was positively associated with contraceptive use in both countries.

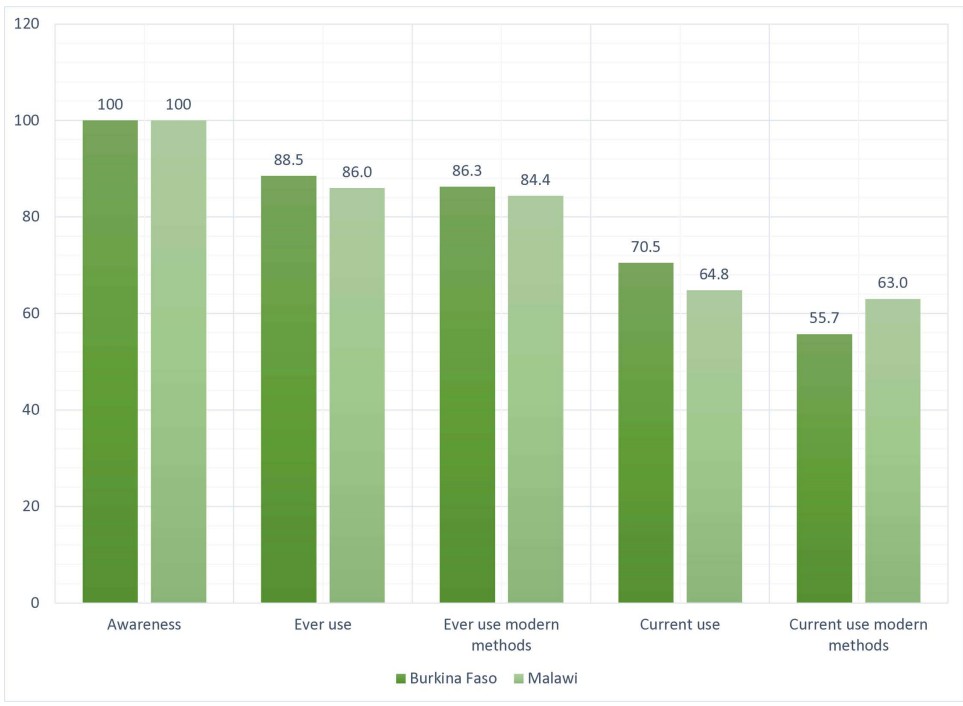

**Fig 1. Contraceptive prevalence (overall).**

**Table 2. Knowledge and use of contraceptives.**

|  | Aware of contraceptives | | Ever used contraceptives | | Current use of contraceptives | |
|---|---|---|---|---|---|---|
|  | B/Faso (N = 628) | Malawi N = 500) | B/Faso (N = 628) | Malawi N = 500) | B/Faso (N = 628) | Malawi N = 500) |
| *Oral Pill* | 561(89.2) | 454(90.8) | 167(27.6) | 57(11.5) | 53(9.3) | 18(3.6) |
| *IUD* | 419(65.9) | 402(80.7) | 51(7.17) | 31(6.2) | 37(5.6) | 14(2.9) |
| *Injectables* | 545(87.1) | 475(94.9) | 207(33.6) | 248(49.4) | 152(24.7) | 208(41.3) |
| *Implant* | 523(83.2) | 428(85.5) | 189(29.8) | 88(17.8) | 71(11.9) | 71(14.4) |
| *Male condom* | 586(93.6) | 456(91.2) | 417(67.4) | 281(56.5) | 141(23.2) | 18(3.7) |
| *Female condom* | 348(57.2) | 376(75.2) | 29(4.87) | 18(3.62) | 53(9.4) | – |
| *Rhythm methods/ Safe days* | 367(58.9) | 338(68.0) | 174(27.3) | 81(16.3) | 45(7.4) | 3(0.6) |
| *Withdrawal* | 273(45.3) | 345(69.2) | 86(15.4) | 137(27.6) | 24(4.1) | 3(0.6) |
| *Emergency contraception* | 258(42.4) | 286(57.6) | 74(13.0) | 20(4.07) | 5(1.1) | 1(0.2.) |

Specifically, the adjusted models show that adolescent mothers aged 19 years were more likely to have used any modern methods (AOR = 1.33, 95%CI 1.03–1.71) or currently use any contraceptive methods (AOR = 1.55, 95% CI 1.07–2.26) compared to those aged 16 or below in Burkina Faso. Similarly, in Malawi, adolescent mothers aged 19 were more likely to have ever used any contraceptive methods (UOR = 1.28 95%CI 1.07–1.53) or currently use any methods (UOR = 1.35 95%CI 1.11–1.63) compared to those aged 16 and below. However, after adjusting for other covariates, the association between age and current contraceptive use and current use of modern methods did not reach a statistically significant level in Malawi.

**Table 3. Correlates of contraceptive awareness and use in Burkina Faso.**

| Factors | Ever use a contraceptive | | | | Current use of contraceptives | | | |
|---|---|---|---|---|---|---|---|---|
| | Any methods | | Modern methods | | Any methods | | Modern methods | |
| | UOR | AOR | UOR | AOR | UOR | AOR | UOR | AOR |
| *Age* | | | | | | | | |
| 16 | Ref | Ref | Ref | Ref | Ref | Ref | Ref | Ref |
| 17 | 1.17(0.91-1.49) | 1.15(0.90-1.48) | 1.25(0.95-1.66) | 1.25(0.95-1.65) | 1.41(0.93-2.13) | 1.40(0.94-2.10) | 1.10(0.68-1.78) | 1.11(0.70-1.78) |
| 18 | 1.23(0.99-1.54) | 1.22(0.97-1.52) | 1.33(1.03-1.72)** | 1.32(1.02-1.70)** | 1.47(1.00-2.14)** | 1.43(0.98-2.09) | 1.25(0.83-1.89) | 1.19(0.79-1.80) |
| 19 | 1.26(1.02-1.57)** | 1.25(1.00-1.56) | 1.34(1.04-1.73)** | 1.33(1.03-1.71)** | 1.59(1.10-2.32)** | 1.55(1.07-2.26)** | 1.34(0.90-2.01) | 1.27(0.85-1.91) |
| *Marital status* | | | | | | | | |
| Married/Co-habiting | Ref | Ref | Ref | Ref | Ref | Ref | Ref | Ref |
| Separated/divorced | 0.92(0.82-1.05) | 0.89(0.78-1.02) | 0.93(0.81-1.06) | 0.90(0.77-1.04) | 0.79(0.63-0.99)** | 0.78(0.60-1.01) | 0.67(0.48-0.93)** | 0.66(0.45-0.96)** |
| Single | 0.92(0.84-1.02) | 0.91(0.81-1.01) | 0.93(0.84-1.02) | 0.92(0.82-1.03) | 0.74(0.61-0.89)*** | 0.75(0.61-0.93)*** | 0.65(0.50-0.84)*** | 0.66(0.49-0.90)*** |
| *Religion* | | | | | | | | |
| Christianity | Ref | Ref | Ref | Ref | Ref | Ref | Ref | Ref |
| Islam | 0.95(0.90-1.00) | 0.94(0.89-0.99)** | 0.94(0.89-1.00)** | 0.93(0.87-0.98)** | 0.91(0.83-1.01) | 0.88(0.80-0.97)** | 0.95(0.83-1.10) | 0.90(0.79-1.04) |
| Other/No religious affiliation | 1.09(1.05-1.14)*** | 1.05(0.98-1.13) | 0.56(0.14-2.23) | 0.53(0.14-2.08) | 1.34(1.25-1.45)*** | 1.24(1.09-1.41)*** | 0.87(0.22-3.50) | 0.77(0.22-2.75) |
| *Worked for pay* | | | | | | | | |
| No | Ref | Ref | Ref | Ref | Ref | Ref | Ref | Ref |
| Yes | 1.02(0.96-1.08) | 1.02(0.97-1.08) | 1.02(0.96-1.09) | 1.02(0.96-1.09) | 1.01(0.91-1.12) | 1.01(0.91-1.12) | 1.04(0.90-1.20) | 1.04(0.90-1.20) |
| *Education* | | | | | | | | |
| Primary | Ref | Ref | Ref | Ref | Ref | Ref | Ref | Ref |
| Secondary | 1.05(0.99-1.11) | 1.05(1.00-1.11) | 1.05(0.98-1.11) | 1.05(0.99-1.12) | 0.99(0.90-1.10) | 1.02(0.92-1.13) | 0.95(0.83-1.10) | 1.00(0.87-1.15) |
| *Living arrangements* | | | | | | | | |
| Not living with any parent | Ref | Ref | Ref | Ref | Ref | Ref | Ref | Ref |
| Living with one parent | 0.99(0.89-1.11) | 1.02(0.91-1.14) | 0.99(0.88-1.12) | 1.00(0.88-1.15) | 0.83(0.65-1.06) | 0.91(0.72-1.16) | 0.72(0.50-1.03) | 0.85(0.59-1.21) |
| living with both parents | 0.96(0.87-1.06) | 1.04(0.92-1.18) | 0.95(0.85-1.07) | 1.03(0.90-1.17) | 0.82(0.67-1.00) | 1.03(0.80-1.31) | 0.74(0.56-0.98)** | 1.05(0.74-1.48) |
| *Residence* | | | | | | | | |
| Rural | Ref | Ref | Ref | Ref | Ref | Ref | Ref | Ref |
| Urban | 1.06(0.99-1.12) | 1.05(0.99-1.12) | 1.06(0.99-1.14) | 1.06(1.00-1.14) | 0.97(0.87-1.08) | 0.98(0.88-1.08) | 1.02(0.89-1.18) | 1.04(0.90-1.20) |

***p<0.001,

**p<0.05, UOR: Unadjusted odds ratio, AOR: Adjusted odds ratio.

Single marital status was significantly associated with lower odds of contraceptive use in both countries. Single adolescent mothers were less likely to currently use any contraceptives (AOR = 0.75, 95%CI 0.61–0.93) and modern methods (AOR = 0.66, 95%CI 0.49–0.90) compared to their married counterparts in Burkina Faso. Similarly, in Malawi, single marital status was negatively associated with having ever used any contraceptive methods (AOR = 0.91, 95% CI 0.83–0.99),

**Table 4. Correlates of contraceptive awareness and use in Malawi.**

| Factors | Ever use a contraceptive | | | | Current use of contraceptives | | | |
|---|---|---|---|---|---|---|---|---|
| | Any methods | | Any modern methods | | Any methods | | Any modern methods | |
| | UOR | AOR | UOR | AOR | UOR | AOR | UOR | AOR |
| *Age* | | | | | | | | |
| 16 | Ref | Ref | Ref | Ref | Ref | Ref | Ref | Ref |
| 17 | 1.12(0.92-1.37) | 1.10(0.90-1.35) | 1.17(0.94-1.45) | 1.14(0.92-1.42) | 1.09(0.79-1.51) | 0.95(0.69-1.32) | 1.18(0.83-1.66) | 1.02(0.73-1.44) |
| 18 | 1.20(1.00-1.45) | 1.17(0.97-1.41) | 1.23 (1.01-1.51)** | 1.20(0.98-1.46) | 1.24(0.92-1.66) | 1.06(0.78-1.43) | 1.28(0.93-1.75) | 1.09(0.79-1.50) |
| 19 | 1.34 (1.13-1.60)*** | 1.28 (1.07-1.53)*** | 1.41 (1.16-1.71)*** | 1.35 (1.11-1.63)*** | 1.52 (1.15-2.00)*** | 1.20(0.90-1.60) | 1.60 (1.18-2.15)*** | 1.25(0.92-1.70) |
| *Marital status* | | | | | | | | |
| Married/ Co-habiting | Ref | Ref | Ref | Ref | Ref | Ref | Ref | Ref |
| Separated/ divorced | 1.05(0.98-1.13) | 1.08(0.99-1.17) | 1.06(0.97-1.14) | 1.09(1.00-1.20) | 0.67 (0.53-0.83)*** | 0.69 (0.54-0.87)*** | 0.61 (0.48-0.78)*** | 0.62(0.48-0.80)*** |
| Single | 0.87 (0.80-0.94)*** | 0.91 (0.83-0.99)** | 0.87 (0.80-0.95)*** | 0.93(0.84-1.02) | 0.54 (0.47-0.63)*** | 0.60 (0.50-0.71)*** | 0.51 (0.44-0.60)*** | 0.55(0.46-0.66)*** |
| *Religion* | | | | | | | | |
| Christianity | Ref | Ref | Ref | Ref | Ref | Ref | Ref | Ref |
| Islam | 1.11 (1.02-1.21)** | 1.13 (1.04-1.23)*** | 1.14 (1.04-1.24)*** | 1.15 (1.05-1.26)*** | 1.19(0.98-1.45) | 1.20 (1.00-1.43)** | 1.23 (1.01-1.50)** | 1.22(1.02-1.47)** |
| Other/No religious affiliation | 1.18 (1.13-1.22)*** | 1.11 (1.04-1.19)*** | 1.20 (1.15-1.25)*** | 1.14 (1.06-1.22)*** | 1.35(0.99-1.84) | 1.21(0.82-1.78) | 1.16(0.72-1.86) | 1.02(0.62-1.69) |
| *Worked for pay* | | | | | | | | |
| No | Ref | Ref | Ref | Ref | Ref | Ref | Ref | Ref |
| Yes | 1.03(0.96-1.11) | 1.00(0.93-1.07) | 1.03(0.95-1.11) | 0.99(0.91-1.07) | 1.11(0.98-1.27) | 1.03(0.91-1.17) | 1.12(0.97-1.28) | 1.04(0.91-1.19) |
| *Education* | | | | | | | | |
| Primary | Ref | Ref | Ref | Ref | Ref | Ref | Ref | Ref |
| Secondary | 1.04(0.97-1.11) | 1.05(0.98-1.13) | 1.04(0.96-1.12) | 1.05(0.97-1.13) | 0.89(0.77-1.03) | 0.96(0.84-1.09) | 0.88(0.76-1.02) | 0.95(0.83-1.08) |
| *Living arrangements* | | | | | | | | |
| Not living with any parent | Ref | Ref | Ref | Ref | Ref | Ref | Ref | Ref |
| Living with one parent | 0.92(0.84-1.01) | 0.96(0.87-1.06) | 0.90 (0.82-0.99)** | 0.94(0.85-1.04) | 0.64 (0.53-0.76)*** | 0.81 (0.67-0.98)** | 0.64 (0.53-0.77)*** | 0.85(0.70-1.04) |
| living with both parents | 0.96(0.88-1.05) | 1.02(0.92-1.12) | 0.96(0.88-1.06) | 1.02(0.92-1.13) | 0.80 (0.67-0.94)*** | 1.06 (0.88-1.28) | 0.80 (0.67-0.95)** | 1.12(0.92-1.37) |
| *Residence* | | | | | | | | |
| Rural | Ref | Ref | Ref | Ref | Ref | Ref | Ref | Ref |
| Urban | 0.97(0.90-1.04) | 0.97(0.90-1.04) | 0.97(0.90-1.04) | 0.97(0.90-1.04) | 0.98(0.86-1.12) | 1.00(0.88-1.13) | 0.99(0.87-1.14) | 1.01(0.89-1.15) |

***p<0.001,

**p<0.05, UOR: Unadjusted odds ratio, AOR: Adjusted odds ratio.

currently using any methods (AOR = 0.60, 95% CI 0.50–0.71) or currently using any modern methods (AOR = 0.55, 95% CI 0.46–0.66).

Adolescent mothers who practice the Islamic religion had lower odds of contraceptive use in Burkina Faso but higher odds in Malawi. More so, adolescent mothers who practice the Islamic religion were six to 12% less likely to use contraceptives

**Table 5. Reasons for non-use of contraceptives.**

| Reason for non-contraceptive use | B/Faso (n = 185) | Malawi (n = 176) |
|---|---|---|
| Infrequent sex/no sex | 65(36.6) | 115(64.9) |
| Religious prohibition | 36(16.6) | 2(1.1) |
| Fear of side effects | 31(17.7) | 17(9.7) |
| Inconvenient to use | 11(6.4) | 2(1.1) |
| Husband/partner opposed | 10(6.1) | 1(0.5) |
| Health concerns | 8(4.3) | 5(2.9) |
| Respondent opposed | 7(3.6) | – |
| Menopausal/had hysterectomy | 6(2.9) | – |
| Wants another child soon | 5(3.5) | 1(0.5) |
| Cost too much | 5(2.6) | – |
| Knows no source | 5(2.6) | 1(0.5) |
| Sub-fecund/infecund | 3(2.1) | – |
| Interfere with the body's normal process | 3(1.7) | 5(2.8) |
| Lack of access/too far | 3(1.7) | 2(1.2) |
| Knows no method | 2(0.9) | 2(1.1) |
| Wants as many children as possible | 1(0.7) | 1(0.5) |
| Others opposed | 1(0.2) | 1(0.5) |

in Burkina Faso (ever use any methods, AOR = 0.94, 95% CI 0.89–0.99; ever used any modern methods, AOR = 0.93, 95% CI 0.87–0.98; current modern method use, AOR = 0.88, 95% CI 0.80–0.97). In contrast, adolescent mothers who practice Islamic religion were more likely to use contraceptives in Malawi (Ever use any methods, AOR = 1.13, 95% CI 1.04–1.23; ever used any modern methods, AOR = 1.15, 95% CI 1.05–1.26; currently use any methods, AOR = 1.20, 95% CI 1.001–1.43; currently use modern methods, AOR = 1.22, 95% CI 1.02–1.47). Residence, education, and employment history were not significantly associated with contraceptive use among adolescent mothers in both countries.

In Table 5, we present reasons for the non-use of contraceptives among the adolescent girls in Burkina Faso and Malawi. For Burkina Faso (n = 185) the most common reason for non-contraceptive use was "infrequent sex/no sex," cited by 37% of respondents. Other reasons for non-use include "religious prohibition" (17%), "fear of side effects" (18%), "health concerns" (4%), "inconvenient to use" (6%), "cost too much" (3%), and "interfere with body's normal process" (2%). The most common reason for non-contraceptive use in Malawi was "Infrequent sex/no sex," cited by a higher percentage of respondents (65%). Other reasons for non-use in Malawi include "fear of side effects" (10%), "health concerns" (3%), "inconvenient to use" (1%), and "interfere with body's normal process" (3%).

## Discussion

In this study, we examined the level of the knowledge, correlates of contraceptive use, and reasons for non-use among adolescent mothers in Burkina Faso and Malawi. The results show that the levels of contraceptive awareness and use were relatively high among adolescent mothers in both Burkina Faso and Malawi. A higher proportion of adolescent mothers in both countries were currently using modern methods compared to the national average, which is crucial for the prevention of rapid repeat pregnancy and the realization of their reproductive health rights. A previous study in this regard found similar findings [17].

We found that age and marital status were associated with contraceptive use in both Burkina Faso and Malawi. Younger adolescent mothers had lower odds of ever and currently using contraceptives. This finding is consistent with findings from Ahinkorah and colleagues in 2021 [15,18] and suggests the need for targeted interventions and awareness

campaigns to promote contraception among younger adolescent mothers. Consistent with a previous study, single marital status was significantly associated with lower odds of contraceptive use [18]. This aligns with the notion that women and girls in stable marital unions or civil partnerships may have greater motivation to plan their families [5,19]. Also, married adolescent mothers may experience improved access to contraceptive methods, as spousal support and reduced societal stigma can facilitate utilization [20]. However, the extent of partner support often falls short of anticipated levels, influencing the selection of contraceptive options, such as a preference for injectables over oral contraceptives, due to their longer-acting nature and reduced reliance on daily spousal cooperation. Religious affiliation was notably associated with contraceptive use, particularly in Burkina Faso, where one in six nonusers attributed their nonuse of contraceptives to religious prohibition. Adolescent mothers practicing Islam had lower odds of contraceptive use compared to Christians in Burkina Faso, but higher odds of use in Malawi. The divergent findings underscore that religion does not operate in a vacuum but interacts with cultural systems, gender norms, and healthcare access. Islamic teachings in Burkina Faso often emphasize traditional family values, including early marriage and pronatalist norms (prioritizing childbearing) [21,22]. Conservative interpretations of Islam may discourage contraceptive use, viewing it as interfering with divine will (Qadar) or as promoting "immoral" sexual autonomy outside marriage [23,24]. Whereas some Islamic leaders in Malawi have endorsed contraception for spacing births (consistent with the Quranic principle of maternal and child health). Organizations like the Islamic Medical Association of Malawi actively promote family planning as compatible with faith, emphasizing 'azl (withdrawal), a historically permitted method [25]. In Burkina Faso, Islam's majority status reinforces conservative interpretations tied to cultural identity. In Malawi, Muslims' minority status may foster adaptation to majority-Christian norms or health systems [26]. These contrasts highlight the need for context-specific interventions that engage religious leaders, address structural barriers, and leverage local cultural frameworks to improve reproductive health outcomes.

The fear of side effects and Infrequent sex/no sex came out as significant reasons and predictors of contraceptive non-use among adolescent mothers. Many adolescent mothers expressed concerns about the potential side effects of contraceptives, such as weight gain, mood changes, or long-term fertility issues. These fears can deter them from initiating or continuing contraceptive use. Additionally, adolescent mothers who perceive their sexual activity as infrequent or nonexistent may feel that contraception is unnecessary, further reducing the likelihood of use. This perception often leads to inconsistent contraceptive use, increasing the risk of RPPs when sexual activity resumes unexpectedly [27–29]

Our findings have critical implications for policies and programs aimed at enhancing contraceptive use and preventing rapid repeat pregnancies among adolescent mothers. Although adolescent mothers show high contraceptive awareness, gaps persist regarding specific methods. Program implementers, providers, and policymakers should intensify efforts to raise awareness about emergency contraception, IUDs, female condoms, and rhythm methods to enhance contraceptive options for adolescent mothers. Despite relatively high contraceptive use, nearly half of adolescent mothers in Burkina Faso and more than one in three in Malawi were either using methods more susceptible to human error or not using any contraceptive methods at all. This demonstrates that gaps remain in terms of contraceptive uptake among adolescent mothers in these two countries. This gap can be closed by programming specifically for single adolescent mothers and those who expressed concerns about side effects or reported infrequent sex. Overall, our findings highlight the need for multifaceted interventions that consider the various factors influencing contraceptive choices. Tailored programs should focus on younger adolescent mothers, addressing their specific concerns and needs. Furthermore, engaging religious and community leaders to advocate for family planning within their cultural contexts could help reduce barriers. This study contributes to the broader global efforts to enhance reproductive autonomy and improve maternal and child health outcomes considering the Sustainable Development Goals' emphasis on reproductive health and gender equality.

## Study strengths and limitations

While this study contributes to understanding contraceptive use and its correlates among adolescent mothers in Burkina Faso and Malawi, certain limitations exist. The study's cross-sectional design means we cannot establish causal

relationships between the single marital status, age, and contraceptive use. Longitudinal studies and qualitative research could provide deeper insights into the dynamics of contraceptive decision-making among adolescent mothers. Additionally, the study was conducted in Burkina Faso's capital and Malawi's second-largest city, areas with greater access to contraceptives compared to other regions. This suggests that the prevalence reported in our study may be higher than in other parts of the country and therefore should be considered when interpreting our findings. Future research should investigate the influence of parental figures, particularly mothers, on adolescents' attitudes toward and utilization of contraceptive methods, as well as examine the role played by male partners in shaping these perspectives and behaviors

## Conclusion

In conclusion, this study sheds light on contraceptive knowledge, use and correlates among adolescent mothers in Burkina Faso and Malawi. The findings highlight the need for tailored interventions that consider younger age, single marital status, religion, infrequent sex, and concerns among side effects. By addressing these intersecting factors within their cultural context, public health efforts can further contribute to improved reproductive health outcomes and family planning choices among adolescent mothers in these countries.

## Author contributions

**Conceptualization:** Emmanuel Oloche Otukpa, Alister Munthali, Nathalie Sawadogo, Boniface Ayanbekongshie Ushie, Anthony Idowu Ajayi.

**Data curation:** Emmanuel Oloche Otukpa, Alister Munthali, Nathalie Sawadogo, Anthony Idowu Ajayi.

**Formal analysis:** Emmanuel Oloche Otukpa, Anthony Idowu Ajayi.

**Funding acquisition:** Boniface Ayanbekongshie Ushie.

**Investigation:** Emmanuel Oloche Otukpa, Alister Munthali, Nathalie Sawadogo.

**Methodology:** Emmanuel Oloche Otukpa, Boniface Ayanbekongshie Ushie.

**Project administration:** Nathalie Sawadogo.

**Supervision:** Anthony Idowu Ajayi.

**Writing – original draft:** Emmanuel Oloche Otukpa, Anthony Idowu Ajayi.

**Writing – review & editing:** Emmanuel Oloche Otukpa, Alister Munthali, Nathalie Sawadogo, Boniface Ayanbekongshie Ushie, Anthony Idowu Ajayi.

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
