## [Decision Letter · Decision Letter 0]

Dear Dr. Otukpa,

Thank you for submitting your manuscript to PLOS ONE. After careful consideration, we feel that it has merit but does not fully meet PLOS ONE’s publication criteria as it currently stands. Therefore, we invite you to submit a revised version of the manuscript that addresses the points raised during the review process.

We look forward to receiving your revised manuscript.

Kind regards,

Alfredo Luis Fort, M.D., M.Sc., Ph.D.

Academic Editor

PLOS ONE

Journal Requirements:

 2. Thank you for stating the following financial disclosure: [This research was part of a larger study "Understanding the context of, and addressing sensitive sexual and reproductive rights issues in sub-Saharan Africa: A regional research and advocacy programme." funded through a grant to the African Population and Health Research Center from the Swedish International Development Cooperation Agency (Sida Contribution No. 12103)]. 

Additional Editor Comments:

Your manuscript is well done and edited. However, there are still a few areas where myself and another reviewer have suggested to do some "minor revisions" in order for the article to be ready for publication. Thanks!

Reviewers' comments:

Reviewer's Responses to Questions

**Comments to the Author**

1. Is the manuscript technically sound, and do the data support the conclusions?

Reviewer #1: Yes

2. Has the statistical analysis been performed appropriately and rigorously?

Reviewer #1: Yes

3. Have the authors made all data underlying the findings in their manuscript fully available?

Reviewer #1: Yes

4. Is the manuscript presented in an intelligible fashion and written in standard English?

Reviewer #1: Yes

Reviewer #1: comt

l68- replace would with may or could

l72- remove various

Results: results for 19yrs & < 16 given- readers may wonder re ages 16-18.

Intro:

Ll68-70- doesn’t flow easily for the reader; would reword to clarify what you’re trying to convey

L68- chajust before nge would to may

L81- you refer to ages 15-24- your upper age for adolescence is confusing & seems out of place with rest of paper

L86- provider bias – clarify r/t to subject of this paper-e.g. reluctance to provide contraceptives to adolescents????

LL98-1-101- This sentence should be rephrased, e.g. We sought to conduct a study that would contribute to the broader global efforts….. or moved to beginning of discussion. The end of the intro should begin with a sentence that includes the purpose of this study, e.g. We designed a cross sectional descriptive study to …….

Materials & Methods: should start with a brief summary of the previous study as described in lines 106- 109 …We used data obtained from a larger mixed methods study……. and then continue with …. You may need to provide a brief explanation about using a two-stage stratified sampling approach to survey the adolescent girls in BF & Malawi

LL 119-120 is the purpose of your study and also belongs at the end of the introduction; just before you state that you designed a cross sectional descriptive study to….

Line 121- clarify what you mean by nuanced in this context

Line 129- term enumeration areas is not clear (at least not to me)

L 135- household listing exercise unclear; next line explains a sampling from that you used- how does this differ from the household listing exercise. Would simplify the language for this methodology

LL 139-141- Would use was sufficient but not say based on the study’s research question-; would summarize as expressed at the end of the introduction (you never clearly state a research question; you’re investigating the prevalence and correlates of contraceptive use among parenting adolescent girls in urban & rural settings….. . As above would put the research goal at the end of the intro. You could also Sart with the sentence beginning on line 139, e.g. Based on the formula for calculating the sample size for an observation study………, we estimated that a sample size of 500 adolescent mothers is sufficient….

L 147- Is there a ref for the pretested questionnaire?

L153- Spell out APHRC (first use) African Population and Health Research Center

LL Outcome Variables- Did you provides samples or pictures of the various methods you asked about?

LL173-187- Which variables were considered Community level covariates?

RESULTS:

L201- Where did most of the participants live (in both countries most didn’t live with parents)

Table 4 shows some ever used differences for 18 yos – consider a mention in results

L255- Use Women of Islamic religion (not Islamic religion) had a sisgnificantly lower odds of contraceptive use…….

Discussion:

L 14- Reword first sentence- In this study we examined …..The results showed

L 17- Higher proportion (higher than who/where???)

**Do you want your identity to be public for this peer review?** For information about this choice, including consent withdrawal, please see our Privacy Policy

Reviewer #1: No

---

## [Author Response · Author response to Decision Letter 1]

25 May 2025

We have attached a point by point response to the reviewer and editor comments in the attachment titled "Reviewer-Editor comments"

---

## [Editor Report · Decision Letter 1]

Prevalence and correlates of contraceptive use among parenting adolescents: results from a cross-sectional survey in Ouagadougou, Burkina Faso, and Blantyre, Malawi

PONE-D-25-04697R1

Dear Dr. Otukpa,

We’re pleased to inform you that your manuscript has been judged scientifically suitable for publication and will be formally accepted for publication once it meets all outstanding technical requirements.

Kind regards,

Alfredo Luis Fort, M.D., M.Sc., Ph.D.

Academic Editor

PLOS ONE

Additional Editor Comments (optional):

Thanks to the authors for addressing the issues by reviewers. The manuscript seems ready for publication, except for a few minor errors to be corrected.
---

## [Editor Report · Acceptance letter]

PONE-D-25-04697R1

PLOS ONE

Dear Dr. Otukpa,

I'm pleased to inform you that your manuscript has been deemed suitable for publication in PLOS ONE. Congratulations! Your manuscript is now being handed over to our production team.

Kind regards,

on behalf of

Dr. Alfredo Luis Fort

Academic Editor

PLOS ONE